# CT Field of View Extension Using Combined Channels Extension and Deep Learning Methods

**Éric Fournié**                                        ERIC.FOURNIE@SIEMENS-HEALTHINEERS.COM

**Matthias Baer-Beck**                            MATTHIAS.BAER@SIEMENS-HEALTHINEERS.COM

**Karl Stierstorfer**                        KARL.STIERSTORFER@SIEMENS-HEALTHINEERS.COM

*Siemens Healthcare GmbH, Computed Tomography, Siemensstr. 3, 91301 Forchheim, Germany*

## Abstract

This paper proposes a method to extend the field of view of computed tomography images. In a first step, the field of view is increased by extrapolating linearly the outer channels in the sinogram space. The modified sinogram is then used to reconstruct extended field of view (EFoV) images containing artifacts due to the channels extension. In a second step, those artifacts are reduced by a deep learning network in image space.

The proposed method has been evaluated on a collection of clinical scans. The resulting volumes have been checked for consistency and plausibility and compared to an existing state of the art EFoV method.

**Keywords:** Extended field of view, CT, sinogram, U-Net, artifact reduction.

## 1. Introduction

X-ray computed tomography (CT) reconstructs slices of the scanned structure from the sinograms acquired by rotating a detector array and an X-ray emitting tube around the scanner gantry. One of its fundamental limitation is that only data that is located inside the scan field of view (SFoV) of the CT scanner can be reconstructed without artifacts. The size of the SFoV is determined by the geometrical properties of the CT scanner, i.e. by the size of the detector and by the distance of the X-ray tube and the detector from the isocenter of the CT scanner. Sinograms nonetheless contains information from regions outside of the SFoV. Artificially extending their angular width, for example by padding or extrapolating the channels as in fig. 1, will allow for a partially reconstructed EFoV region polluted by artifacts.

The proposed method first extends the sinograms in channel direction, then uses a U-Net (Ronneberger et al., 2015) to clean up the artifacts. The result is a realistic and computationally cheap extension of the field of view. Compared to existing methods (Hsieh et al., 2004; Kunze et al., 2007; Bruder et al., 2008), the proposed approach delivers improved results at a low computational cost.

## 2. Method

The proposed method consists of a two-step approach. Let us consider a scanned object $y$ and its sinogram obtained by the Radon transform $\Re(y)$. In a first step, the angular width

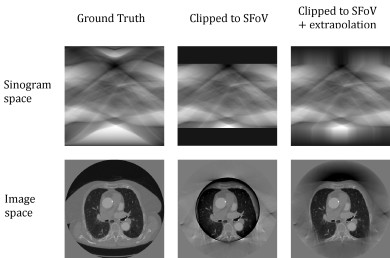

Figure 1: Field of view extension by channel padding or extrapolation.

of the sinogram is increased by linear extrapolation $g(\Re(y))$ of the outermost channels linearly towards zero, thus increasing the number of channels by a factor corresponding to $EFoV/SFoV$. We can now use any method $\Re^+$ such as filtered backprojection (FBP) to reconstruct an EFoV image $X = \Re^+(g(\Re(y)))$ containing artifacts. The SFoV region is only slightly influenced by the extrapolation via the convolution step of the FBP and therefore can be used for diagnostic purposes without limitation.

The artifacts produced outside of the SFoV region in $X$ are then cleaned up by a U-Net $F$ trained to match $(X, y)$ pairs. The output of the network $\hat{y} = F(X)$ can then be used as a realistic estimation of $y$ for the EFoV region. With this approach, the problem faced is the reduction of artifacts instead of inpainting large portions of the image, thus largely reducing the space of acceptable solutions.

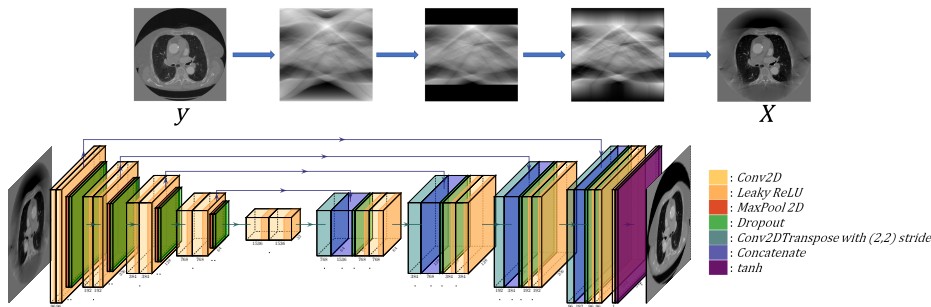

Figure 2: Data generation pipeline and network architecture

To remediate possible modifications of the SFoV region by the network, it is replaced by the corresponding SFoV of $X$. In order to improve the final SFoV/EFoV transition, the network output is adapted to the input distribution:

$$\hat{y} = \frac{\hat{y} - \mu_{\hat{y}}}{\sigma_{\hat{y}}} \cdot \sigma_X + \mu_X$$

To train the network, a collection of clinical CT scan slices $y$ are projected to their Radon transform to obtain sinograms. The sinograms are extended by extrapolation of the outer channels as described previously (cf. fig. 1 and 2) and backprojected to images $X$

used as inputs of the network. The loss function is only computed in the EFoV region and is defined as the weighted sum of the structural dissimilarity $DSSIM = (1 - SSIM)/2$ (Wang et al., 2004) and the mean squared error.

## 3. Results and Conclusions

To evaluate performance on clinical data, a plugin implementing the various steps of the method including execution of the trained network has been developed for ReconCT, a proprietary offline reconstruction software developed by Siemens Healthineers. ReconCT can also reconstruct EFoV images using the existing state of the art HDFoV method.

Scanner raw data corresponding to 11 patients and 4 phantoms has been collected and EFoV images reconstructed using the proposed method and HDFoV. While no ground truth exists for the patient data sets, reconstructions have been checked for artifacts errors and plausibility in axial, coronal and sagittal slices. Some qualitative results are presented in fig. 3. When comparing the results depicted in figure 3 it shows up that the results with the proposed method are superior to the HDFoV results for cases 2, 4 and 5. For cases 1 and 3 the proposed method cannot outperform the HDFoV method as it either introduces artificial structures (case 1) or fails to reconstruct some parts of the patient anatomy (arm in case 3).

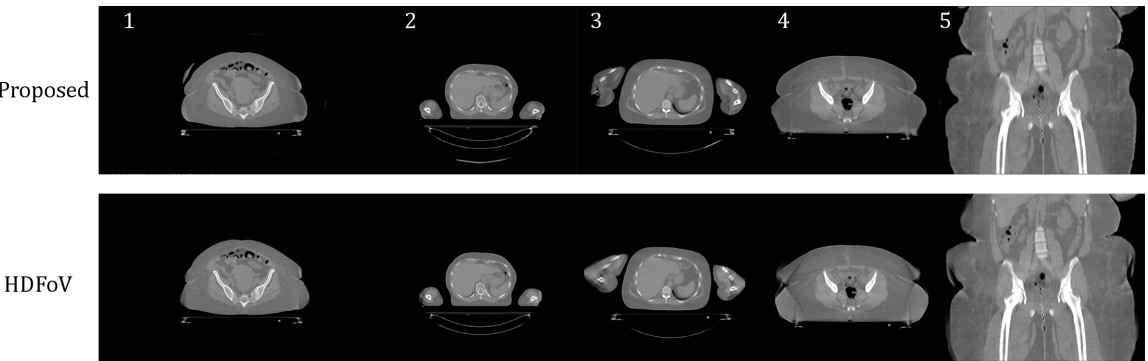

Figure 3: Comparison between our method (top) and HDFoV (bottom)

Extrapolation by linear extrapolation followed by the removal of artifacts in the corresponding reconstruction using a deep learning network proved to be an efficient method to extend the field of view of CT scans and shows potential to improve quality of EFoV reconstructions. Like all methods aiming for the reconstruction of volume areas using incomplete projection data, the EFoV regions cannot be used for diagnostic but they are sufficient for an array of clinical applications where the normal SFoV is too narrow, such as radio therapy planning or the imaging of obese patients.

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
