# OpenReview forum: "CT Field of View Extension Using Combined Channels Extension and Deep Learning Methods"
_MIDL.io/2019/Conference/Abstract — MIDL Abstract 2019_

### Official Review · AnonReviewer1 · 2019-04-28
**U-net for extrapolation artifact removal**

**Rating:** 3
**Confidence:** 1

**Review:**

They train a standard U-net to learn to clean up artifacts after extrapolation of the field of view. The contribution is that they pose an inpainting task as an artifact-removal, which is easier. The method is compared to a previous state of the art method and qualitatively shows improvements

---

### Official Review · AnonReviewer2 · 2019-05-01
**promising method; no quantitative validation**

**Rating:** 3
**Confidence:** 1

**Review:**

This study addresses the important problem of extending the FOV in CT. The authors’ proposed algorithm is compared against an existing state-of-the art method. Yet, as no quantitative validation is provided (only qualitative, where it performs better in some cases but not others), it's hard to assess the benefits of this approach over existing work.
While the authors mention that no ground truth exists in patient data, I wondered if quantitative results be obtained by reducing the actual number of channels used for the reconstruction, then applying the authors’ approach, and comparing the resulting reconstruction to that obtained with the full number of channels?

---

### Decision · Program_Chairs · 2019-05-06
**Acceptance Decision**

Accept